# Nonparametric learning from Bayesian models with randomized objective functions

**Simon Lyddon**
Department of Statistics
University of Oxford
Oxford, UK
lyddon@stats.ox.ac.uk

**Stephen Walker**
Department of Mathematics
University of Texas at Austin
Austin, TX
s.g.walker@math.utexas.edu

**Chris Holmes**
Department of Statistics
University of Oxford
Oxford, UK
cholmes@stats.ox.ac.uk

## Abstract

Bayesian learning is built on an assumption that the model space contains a true reflection of the data generating mechanism. This assumption is problematic, particularly in complex data environments. Here we present a Bayesian nonparametric approach to learning that makes use of statistical models, but does not assume that the model is true. Our approach has provably better properties than using a parametric model and admits a Monte Carlo sampling scheme that can afford massive scalability on modern computer architectures. The model-based aspect of learning is particularly attractive for regularizing nonparametric inference when the sample size is small, and also for correcting approximate approaches such as variational Bayes (VB). We demonstrate the approach on a number of examples including VB classifiers and Bayesian random forests.

## 1   Introduction

Bayesian updating provides a principled and coherent approach to inference for probabilistic models [23], but is predicated on the model class being true. That is, for an observation $x$ and a generative model $F_\theta(x)$ parametrized by a finite-dimensional parameter $\theta \in \Theta$, then for some parameter value $\theta_0 \in \Theta$ it is that $x \sim F_{\theta_0}(x)$. In reality, however, all models are false. If the data is simple and small, and the model space is sufficiently rich, then the consequences of model misspecification may not be severe. However, data is increasingly being captured at scale, both in terms of the number of observations as well as the diversity of data modalities. This poses a risk in conditioning on an assumption that the model is true.

In this paper we discuss a scalable approach to Bayesian nonparametric learning (NPL) from models without the assumption that $x \sim F_{\theta_0}(x)$. To do this we use a nonparametric prior that is centered on a model but does not assume the model to be true. A concentration parameter, $c$, in the nonparametric prior quantifies trust in the baseline model and this is subsequently reflected in the nonparametric update, through the relative influence given to the model-based inference for $\theta$. In particular, $c \to \infty$ recovers the standard model-based Bayesian update while $c = 0$ leads to a Bayesian bootstrap estimator for the object of interest.

Our methodology can be applied in a number of situations, including:

[S1] Model misspecification: where we have used a parametric Bayesian model and we are concerned that the model may be misspecified.

[S2] Approximate posteriors: where for expediency we have used an approximate posterior, such as in variational Bayes (VB), and we wish to account for the approximation.

[S3] Direct updating from utility-functions: where the sole purpose of the modelling task is to perform some action or take a decision under a well-specified utility function.

Our work builds upon previous ideas including [21] who introduced the weighted likelihood bootstrap (WLB) as a way of generating approximate samples from the posterior of a well-specified Bayesian model. [19] highlighted that the WLB in fact provides an exact representation of uncertainty for the model parameters that minimize the Kullback-Leibler (KL) divergence, $d_{\mathrm{KL}}(F_0, F_\theta)$, between the unknown data-generating distribution and the model likelihood $f_\theta(x)$, and hence is well motivated regardless of model validity. These approaches however do not allow for the inclusion of prior knowledge and do not provide a Bayesian update as we do here.

A major underlying theme behind our paper, and indeed an open field for future research, is the idea of obtaining targeted posterior samples via the maximization of a suitably randomized objective function. The WLB randomizes the log-likelihood function, effectively providing samples which are randomized maximum likelihood estimates, whereas we randomize a more general objective function under a Bayesian nonparametric (NP) posterior. The randomization takes into account knowledge captured through the choice of a model and parametric prior.

## 2  Foundations of Nonparametric Learning

We begin with the simplest scenario, namely [S1], concerning a possibly misspecified model before moving on to more complicated situations. It is interesting to note that all of what follows can also be considered from a viewpoint of NP regularization, using a parametric model to centre a Bayesian NP analysis in a way that induces stability and parametric structure to the problem.

### 2.1  Bayesian updating of misspecified models

Suppose we have a parametric statistical model, $\mathcal{F}_\Theta = \{f_\theta(\cdot); \ \theta \in \Theta)\}$, where for each $\theta \in \Theta \subseteq \mathbb{R}^p$, $f_\theta : \mathcal{X} \to \mathbb{R}$ is a probability density. The conventional approach to Bayesian learning involves updating a prior distribution to a posterior through Bayes' theorem. This approach is well studied and well understood [3], but formally assumes that the model space contains the true data-generating mechanism. We will derive a posterior update under weaker assumptions.

Suppose that $\mathcal{F}_\Theta$ has been selected for the purpose of a prediction, or a decision, or some other modelling task. Consider the thought experiment where the modeller somehow gains access to Nature's true sampling distribution for the data, $F_0(x)$, which does not necessarily belong to $\mathcal{F}_\Theta$. How should they then update their model?

With access to $F_0$ the modeller can simply request an infinite training set, $x_{1:\infty} \overset{iid}{\sim} F_0$, and then update to the posterior $\pi(\theta|x_{1:\infty})$. Under an infinite sample size all uncertainty is removed and for regular models the posterior concentrates at a point mass at $\theta_0$, the parameter value maximizing the expected log-likelihood, assuming that the prior has support there; i.e.

$$\theta_0 = \underset{\theta \in \Theta}{\arg\max} \ \lim_{n \to \infty} n^{-1} \sum_{i=1}^{n} \log f_\theta(x_i) = \underset{\theta \in \Theta}{\arg\max} \int_{\mathcal{X}} \log f_\theta(x) \, dF_0(x).$$

It is straightforward to see that $\theta_0$ minimizes the KL divergence from the true data-generating mechanism to a density in $\mathcal{F}_\Theta$

$$\theta_0 = \underset{\theta \in \Theta}{\arg\max} \int_{\mathcal{X}} \log f_\theta(x) dF_0(x) = \underset{\theta \in \Theta}{\arg\min} \int_{\mathcal{X}} \log \frac{f_0(x)}{f_\theta(x)} dF_0(x). \tag{1}$$

This is true regardless of whether $F_0$ is in the model space of $\mathcal{F}_\Theta$ and is well-motivated as the target of statistical model fitting [1, 8, 27, 5].

Uncertainty in this unknown value $\theta_0$ flows directly from uncertainty in $F_0$. Of course $F_0$ is unknown, but being "Bayesian" we can place a prior on it, $\pi(F)$, for $F \in \mathcal{F}$, that should reflect our honest uncertainty about $F_0$. Typically the prior should have broad support unless we have special knowledge to hand, which is a problem with a parametric modelling approach that only supports a family of distribution functions indexed by a finite-dimensional parameter. The Bayesian NP literature however provides a range of priors for this sole purpose [14]. Once a prior for $F$ is chosen, the correct way to propagate uncertainty about $\theta$ comes naturally from the posterior distribution for the law $\mathcal{L}[\theta(F)|x_{1:n}]$, via $\mathcal{L}[F|x_{1:n}]$, where $\theta(F) = \arg\max_{\theta \in \Theta} \int \log f_\theta(x) dF(x)$. The posterior for the parameter is then captured in the marginal by treating $F$ as a latent auxiliary probability measure,

$$\tilde{\pi}(\theta \mid x_{1:n}) = \int_{\mathcal{F}} \pi(\theta, dF \mid x_{1:n}) = \int_{\mathcal{F}} \pi(\theta \mid F)\pi(dF \mid x_{1:n}), \tag{2}$$

where $\pi(\theta|F)$ assigns probability 1 to $\theta = \theta(F)$. We use $\tilde{\pi}$ to denote the NP update to distinguish it from the conventional Bayesian posterior $\pi(\theta|x_{1:n}) \propto \pi(\theta) \prod_{i=1}^{n} f_\theta(x_i)$, noting that in general the nonparametric posterior $\tilde{\pi}(\theta \mid x_{1:n})$ will be different to the standard Bayesian update as they are conditioning on different states of prior knowledge. In particular, as stated above, $\pi(\theta|x_{1:n})$ assumes artificially that $F_0 \in \mathcal{F}_\Theta$.

## 2.2 An NP prior using a MDP

For our purposes, the mixture of Dirichlet processes (MDP) [2] is a convenient vehicle for specifying prior beliefs $\pi(F)$ centered on parametric models.[1] The MDP prior can be written as

$$[F \mid \theta] \sim \mathrm{DP}(c, f_\theta(\cdot)); \qquad \theta \sim \pi(\theta). \tag{3}$$

This is a mixture of standard Dirichlet processes with mixing distribution or hyper-prior $\pi(\theta)$, and concentration parameter $c$. We write this as $F \sim \mathrm{MDP}(\pi(\theta), c, f_\theta(\cdot))$.

The MDP provides a practical, simple posterior update. From the conjugacy property of the DP applied to (3), we have the conditional posterior update given data $x_{1:n}$, as

$$[F \mid \theta, x_{1:n}] \sim \mathrm{DP}\left(c + n, \ \frac{c}{c+n}f_\theta(\cdot) + \frac{1}{c+n}\sum_{i=1}^{n}\delta_{x_i}(\cdot)\right) \tag{4}$$

where $\delta_x$ denotes the Dirac measure at $x$. The concentration parameter $c$ is an effective sample size, governing the trust we have in $f_\theta(x)$. The marginal posterior distribution for $\mathcal{L}[F|x_{1:n}]$ can be written as

$$\pi(dF \mid x_{1:n}) = \int_\Theta \pi(dF \mid \theta, x_{1:n})\,\pi(\theta \mid x_{1:n})\,d\theta, \tag{5}$$

i.e.

$$[F \mid x_{1:n}] \sim \mathrm{MDP}\left(\pi(\theta \mid x_{1:n}), \ c+n, \ \frac{c}{c+n}f_\theta(\cdot) + \frac{1}{c+n}\sum_{i=1}^{n}\delta_{x_i}(\cdot)\right). \tag{6}$$

The mixing distribution $\pi(\theta|x_{1:n})$ coincides with the parametric Bayesian posterior, $\pi(\theta|x_{1:n})$, assuming there are no ties in the data [2], although as noted above it does not follow that the NP marginal $\tilde{\pi}(\theta|x_{1:n})$ is equivalent to the parametric Bayesian posterior $\pi(\theta|x_{1:n})$.

We can see from the form of the conditional MDP (4) that the sampling distribution of the centering model, $f_\theta(x)$, regularizes the influence of the empirical data $\sum_{i=1}^{n}\delta_{x_i}(\cdot)$. The resulting NP posterior (5) combines the information from the posterior distribution of the centering model $\pi(\theta|x_{1:n})$ with the information in the empirical distribution of the data. This leads to a simple and highly parallelizable Monte Carlo sampling scheme as shown below.

## 2.3 Monte Carlo conditional maximization

The marginal in (2) facilitates a Monte Carlo estimator for functionals of interest under the posterior, which we write as $G = \int_\Theta g(\theta)\tilde{\pi}(\theta|x_{1:n})d\theta$. This is achieved by sampling $\pi(\theta, dF|x_{1:n})$ jointly

from the posterior,

$$\int_\Theta g(\theta)\tilde{\pi}(\theta \mid x_{1:n})d\theta \quad \approx \quad \frac{1}{B}\sum_{i=1}^{B}g(\theta^{(i)})$$

$$\theta^{(i)} = \theta(F^{(i)}) = \arg\max_{\theta\in\Theta}\int_\mathcal{X}\log f_\theta(x)dF^{(i)}(x) \tag{7}$$

$$F^{(i)} \sim \pi(dF \mid x_{1:n}). \tag{8}$$

This involves an independent Monte Carlo draw (8) from the MDP marginal followed by a conditional maximization of an objective function (7) to obtain each $\theta^{(i)}$. This Monte Carlo conditional maximization (MCCM) sampler is highly amenable to fast implementation on distributed computer architectures; given the parametric posterior samples, each NP posterior sample, $F^{(i)}$, can be computed independently and in parallel from (8).

We can see from (6) that the parametric posterior samples are not required if $c = 0$. If $c > 0$ it may be computationally intensive to generate samples from the parametric posterior. However, as we will see next, we do need to sample from this posterior directly. This makes the approach particularly attractive to fast, tractable approximations for $\pi(\theta|x_{1:n})$, such as a variational Bayes (VB) posterior approximation. The NP update corrects for the approximation in a computationally efficient manner, leading to a posterior distribution with optimal properties as shown below.

## 2.4 A more general construction

So far we have assumed, hypothetically, that:

(i) the modeller is interested in learning about the MLE under an infinite sample size, $\theta_0 = \arg\max_\theta \int \log f_\theta(x)dF_0(x)$, rather than $\alpha_0 = \arg\max_\alpha \int u(x,\alpha)dF_0(x)$ more generally, for a utility function $u(x,\alpha)$.

(ii) the parametric mixing distribution $\pi(\theta|x_{1:n})$ of the MDP posterior in (6) is constructed from the same centering model that defines the target parameter, $\theta_0 = \arg\max_\theta \int \log f_\theta(x)dF_0(x)$.

Both of these assumptions can be relaxed. For the latter case, it is valid to use a tractable parametric mixing distribution $\pi(\gamma|x_{1:n})$ and baseline model $f_\gamma$, while still learning about $\theta_0$ in (1) through the marginal $\tilde{\pi}(\theta|x_{1:n})$ as in (2) obtained via $\theta(F)$ and

$$[F \mid x_{1:n}] \sim \text{MDP}\left(\pi(\gamma \mid x_{1:n}),\ c+n,\ \frac{c}{c+n}f_\gamma(\cdot) + \frac{1}{c+n}\sum_i \delta_{x_i}\right). \tag{9}$$

For (i), we can use the mapping $\alpha(F) = \arg\max_\alpha \int u(x,\alpha)dF(x)$ to derive the NPL posterior on actions or parameters maximizing some expected utility under a model-centered MDP posterior. This can be written as $\tilde{\pi}(\alpha|x_{1:n}) = \int \pi(\alpha|F)\pi(dF|x_{1:n})$, where $\pi(\alpha|F)$ assigns probability 1 to $\alpha = \alpha(F)$.

This highlights a major theme of the paper: the idea of obtaining posterior samples via maximization of a suitably randomized objective function. In generality the target is $\alpha_0 = \arg\max_\alpha \int u(x,\alpha)dF_0(x)$, obtained by maximizing an objective function, and the randomization arises from the uncertainty in $F_0$ through $\pi(F|x_{1:n})$ that takes into account the information, and any misspecification, associated with a parametric centering model.

## 2.5 The Posterior Bootstrap algorithm

We will use the general construction of Section 2.4 to describe a sampling algorithm. We assume we have access to samples from the posterior parametric mixing distribution, $\pi(\gamma|x_{1:n})$, in the MDP. In the case of model misspecification, [S1], if the data contains no ties, this is simply the parametric Bayesian posterior under $\{f_\gamma(x), \pi(\gamma)\}$, for which there is a large literature of computational methods available for sampling - see for example [24]. If there are ties then we refer the reader to [2] or note that we can simply break ties by adding a new pseudo-variable, such as $x^* \sim N(0, \epsilon^2)$ for small $\epsilon$.

The sampling algorithm, found in Algorithm 1, is a mixture of Bayesian posterior bootstraps. After a sample $\gamma^{(i)}$ is drawn from the mixing posterior, $\pi(\gamma|x_{1:n})$, a posterior pseudo-sample is generated,

---

**Algorithm 1:** The Posterior Bootstrap

---

**Data:** Dataset $x_{1:n} = (x_1, \ldots, x_n)$.
Parameter of interest $\alpha_0 = \alpha(F_0) = \arg\max_\alpha \int u(x, \alpha) dF_0(x)$.
Mixing posterior $\pi(\gamma|x_{1:n})$, concentration parameter $c$, centering model $f_\gamma(x)$.
Number of centering model samples $T$.
**begin**
    **for** $i = 1, \ldots, B$ **do**
        Draw centering model parameter $\gamma^{(i)} \sim \pi(\gamma|x_{1:n})$;
        Draw posterior pseudo-sample $x^{(i)}_{(n+1):(n+T)} \overset{iid}{\sim} f_{\gamma^{(i)}}$;
        Generate weights
        $(w_1^{(i)}, \ldots, w_n^{(i)}, w_{n+1}^{(i)}, \ldots, w_{n+T}^{(i)}) \sim \text{Dirichlet}(1, \ldots, 1, c/T, \ldots, c/T)$;
        Compute parameter update

$$\tilde{\alpha}^{(i)} = \arg\max_\alpha \left\{ \sum_{j=1}^{n} w_j^{(i)} u(x_j, \alpha) + \sum_{j=1}^{T} w_{n+j}^{(i)} u(x_{n+j}^{(i)}, \alpha) \right\};$$

    **end**
    Return NP posterior sample $\{\tilde{\alpha}^{(i)}\}_{i=1}^{B}$.
**end**

---

$x^{(i)}_{(n+1):(n+T)} \overset{iid}{\sim} f_{\gamma^{(i)}}(x)$, and added to the dataset, which is then randomly weighted. The parameter under this implicit distribution function is then computed as the solution of an optimization problem.

Note for the special case of correcting model misspecification [S1], we have $\gamma \equiv \theta$, $f_\gamma(\cdot) \equiv f_\theta(\cdot)$, $\pi(\gamma|x_{1:n}) \equiv \pi(\theta|x_{1:n})$, $\alpha \equiv \theta$, $u(x, \alpha) \equiv \log f_\theta(x)$, so that the posterior sample is given by

$$\tilde{\theta}^{(i)} = \arg\max_\theta \left\{ \sum_{j=1}^{n} w_j^{(i)} \log f_\theta(x_j) + \sum_{j=1}^{T} w_{n+j}^{(i)} \log f_\theta(x_{n+j}^{(i)}) \right\}.$$

where $w^{(i)} \sim \text{Dirichlet}(\cdot)$ following Algorithm 1 and $x^{(i)}_{(n+1):(n+T)}$ are $T$ synthetic observations drawn from the parametric sampling distribution under $\theta^{(i)}$ which itself is drawn from $\pi(\theta|x_{1:n})$. We leave the concentration parameter $c$ to be set subjectively by the practitioner, representing faith in the parametric model. Some further guidance to the setting of $c$ can be found in Section 1 of the Supplementary Material.

## 2.6 Adaptive Nonparametric Learning: aNPL

Instead of the Dirichlet distribution approximation to the Dirichlet process, we propose an alternative stick-breaking procedure that has some desirable properties. This procedure entails following the usual DP stick-breaking construction [25] for the model component of the MDP posterior, by repeatedly drawing $\text{Beta}(1, c)$-distributed stick breaks, but terminating when the unaccounted for probability measure $\prod_j (1 - v_j)$, multiplied by the average mass assigned to the model, $c/(c + n)$, drops below some threshold $\epsilon$ set by the user. This adaptive nonparametric learning (aNPL) algorithm is written out in full in Section 2 of the Supplementary Material.

One advantage of this approach is that a number of theoretical results then hold, as for large enough $n$, under this adaptive scheme the parametric model is in effect 'switched off', and essentially the MDP with $c = 0$ is used to generate posterior samples. This is an interesting notion in itself. For small samples, we prefer the regularization that our model provides, though as $n$ grows the average probability mass assigned to the model decays like $(c + n)^{-1}$, as seen in (4). In the adaptive version, we agree a hard threshold at which point we discard the model entirely and allow the data to speak for itself. We set this point at a level such that we are a priori comfortable that there is enough information in the sample alone with which to quantify uncertainty in our parameter of interest. For example, $\epsilon = 10^{-4}$ and $c = 1$ only utilizes the centering model for $n < 10,000$. Further, we could use this idea to set $c$: this quantity is determined if a tolerance level, $\epsilon$, and a threshold $n_{\max}$ over which the parametric model would be discarded, are provided by the practitioner.

## 2.7 Properties of NPL

Bayesian nonparametric learning has a number of important properties that we shall now describe.

**Honesty about correctness of model.** Uncertainty in the data-generating mechanism is quantified via a NP update that takes into account the model likelihood, prior, and concentration parameter $c$. Uncertainty about model parameters flows from uncertainty in the data-generating mechanism.
**Incorporation of prior information.** The prior for $\theta$ is naturally incorporated as a mixing distribution for the MDP. This is in contrast to a number of Bayesian methods with similar computational properties but that do not admit a prior [21, 9].
**Parallelized bootstrap computation.** As shown in Section 2.5, NPL is trivially parallelizable through a Bayesian posterior bootstrap and can be coupled with misspecified models or approximate posteriors to deliver highly scalable and exact inference.
**Consistency.** Under mild regularity, all posterior mass concentrates in any neighbourhood of $\theta_0$ as defined in (1), as the number of observations tends to infinity. This follows from an analogous property of the DP (see, for example [14]).
**Standard Bayesian inference is recovered as $c \to \infty$.** This follows from the property of the DP that it converges to the prior degenerate at the base probability distribution in the limit of $c \to \infty$.
**Non-informative learning with $c = 0$.** If no model or prior is available, setting $c = 0$ recovers the WLB. This has an exact interpretation as an objective NP posterior [19], where the asymptotic properties of the misspecified WLB were studied. [20] demonstrated the suboptimality of a misspecified Bayesian posterior, asymptotically, relative to an asymptotic normal distribution with the same centering but a sandwich covariance matrix [15]. We will see next that for large samples the misspecified Bayesian posterior distribution is predictively suboptimal as well.
**A superior asymptotic uncertainty quantification to Bayesian updating.** A natural way to compare posterior distributions is by measuring their predictive risk, defined as the expected KL divergence of the posterior predictive to $F_0$. We consider only the situation where there is an absence of strong prior information, following [26, 12].

We say predictive $\pi_1$ asymptotically dominates $\pi_2$ up to $o(n^{-k})$ if for all distributions $q$ there exists a non-negative and possibly positive real-valued functional $K(q(\cdot))$ such that:

$$\mathbb{E}_{x_{1:n} \sim q} \, d_{\mathrm{KL}}(q(\cdot), \pi_2(\,\cdot\mid x_{1:n})) - \mathbb{E}_{x_{1:n} \sim q} \, d_{\mathrm{KL}}(q(\cdot), \pi_1(\,\cdot\mid x_{1:n})) \;=\; K(q(\cdot)) + o(n^{-k}).$$

We have the following theorem about the asymptotic properties of the MDP with $c = 0$. This result holds for aNPL, as the model component is ignored for suitably large $n$.

**Theorem 1.** *The posterior predictive of the MDP with $c = 0$ asymptotically dominates the standard Bayesian posterior predictive up to $o(n^{-1})$.*

*Proof.* In [12] the bootstrap predictive is shown to asymptotically dominate the standard Bayesian predictive up to $o(n^{-1})$. In Theorem 1 of [13], the predictive of the MDP with $c = 0$ and the bootstrap predictive are shown to be equal up to $o_p(n^{-3/2})$. A Taylor expansion argument shows that the predictive risk of the MDP with $c = 0$ has the same asymptotic expansion up to $o(n^{-1})$ as that of the bootstrap. Thus Theorem 2 of [12] can be proven with the predictive of the MDP with $c = 0$ in place of the bootstrap predictive. Thus the predictive of the MDP with $c = 0$ must also dominate the standard Bayesian predictive up to $o(n^{-1})$. $\square$

## 3 Illustrations

### 3.1 Exponential family, [S1]

Suppose the centering model is an exponential family with parameter $\theta$ and sufficient statistic $s(x)$,

$$\mathcal{F}_\Theta = \left\{ f_\theta(x) = g(x) \exp\left\{ \theta^T s(x) - K(\theta) \right\} ; \ \theta \in \Theta \right\}.$$

Under assumed regularity, by differentiating under the integral sign of (1) we find that our parameter of interest must satisfy $\mathbb{E}_{F_0} s(x) = \nabla_\theta K(\theta_0)$. For a particular $F$ drawn from the posterior bootstrap, the expected sufficient statistic is

$$\nabla_\theta K(\tilde{\theta}) = \lim_{T \to \infty} \left\{ \sum_{j=1}^{n} w_j s(x_j) + \sum_{j=n+1}^{n+T} w_j s(x_j) \right\}.$$

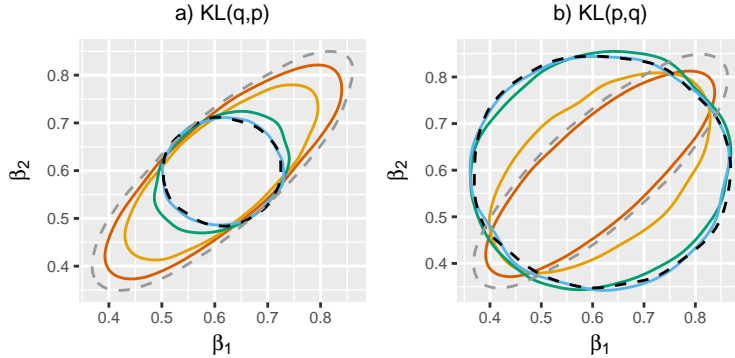

Figure 1: Posterior 95% probability contour for a bivariate Gaussian, comparing VB-NPL with $c \in \{1, 10^2, 10^3, 10^4\}$ (red, orange, green, blue respectively) to the known Bayes posterior (grey dashed line) and a VB approximation (black dashed line).

with $\tilde{\theta}$ the NP posterior parameter value, weights $w_{1:(n+T)}$ arising from the Dirichlet distribution as set out in Algorithm 1, and $x_j \sim f_\theta(\cdot)$ for $j = n+1, \ldots, n+T$, with $\theta$ drawn from the parametric posterior. This provides a simple geometric interpretation of our method, as convex combinations of (randomly-weighted) empirical sufficient statistics and model sufficient statistics from the parametric posterior. The distribution of the random weights is governed by $c$ and $n$ only. Our method generates stochastic maps from misspecified posterior samples to corrected NP posterior samples, by incorporating information in the data over and above that captured by the model.

### 3.2   Updating approximate posteriors [S2]: Variational Bayes uncertainty correction

Variational approximations to Bayesian posteriors are a popular tool for obtaining fast, scalable but approximate Bayesian posterior distributions [4, 6]. The approximate nature of the variational update can be accounted for using our approach. Figure 1 shows a mean-field normal approximation $q$ to a correlated normal posterior $p$, an example similar to one from [4], Section 10.1. We generated 100 observations from a bivariate normal distribution, centered at $(\frac{1}{2}, \frac{1}{2})$, with dimension-wise variances both equal to 1 and correlation equal to 0.9, and independent normal priors on each dimension, both centered at 0 with variance $10^2$. Each posterior contour plotted is based on $10,000$ posterior samples.

By applying the posterior bootstrap with a VB posterior (VB-NPL) in place of the Bayesian posterior, we recover the correct covariance structure for decreasing prior concentration $c$. If instead of $d_{\mathrm{KL}}(q, p)$ we use $d_{\mathrm{KL}}(p, q)$ as the objective, as it is for expectation propagation, the model posterior uncertainty may be overestimated, but is still corrected by the posterior bootstrap.

We demonstrate this in practice through a VB logistic regression model fit to the Statlog German Credit dataset, containing 1000 observations and 25 covariates (including intercept), from the UCI ML repository [10], preprocessing via [11]. An independent normal prior with variance 100 was assigned to each covariate, and 1000 posterior samples were generated for each method. We obtain a mean-field VB sample using automatic differentiation variational inference (ADVI) in Stan [17]. When generating synthetic samples for the posterior bootstrap, both features and classes are needed. Classes are generated, given features, according to the probability specified by the logistic distribution. In this example (and the example in Section 3.3) we repeatedly re-use the features of the dataset for our pseudo-samples. In Fig. 2 we show that the NP update effectively corrects the VB approximation for small values of $c$. Of course, local variational methods can provide more accurate posterior approximations to Bayesian logistic posteriors [16], though these too are approximations, that NP updating can correct.

**Comparison with Bayesian logistic regression.** The conventional Bayesian logistic regression assumes the true log-odds of each class is linear in the predictors, and would use MCMC for inference [22]. The MCMC samples, shown as points in Fig. 2, are a good match to the NPL update but MCMC requires a user-defined burn-in and convergence checking. The runtime to generate 1 million samples by MCMC (discarding an equivalent burn-in), was 33 minutes, compared to 21 seconds with

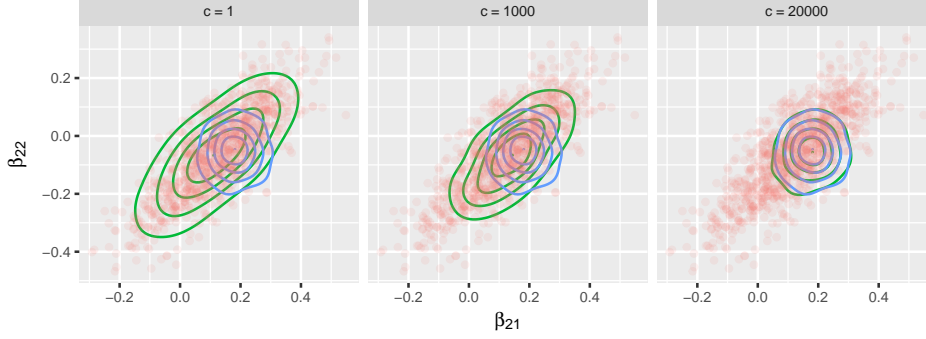

Figure 2: Posterior contour plot for $\beta_{22}$ vs $\beta_{21}$, for VB-NPL (green) and VB (blue), for three values of $c$. Scatter plot is a Bayesian logistic posterior sample (red) via a Polya-Gamma scheme.

NPL, using an m5.24xlarge AWS instance with 96 vCPUs; a speed-up of 95 times. Additionally NPL has provably better predictive properties, as detailed in Section 2.7.

### 3.3 Directly updating the prior: Bayesian Random Forests, using synthetic generated data

Random forests (RF) [7] is an ensemble learning method that is widely used and has demonstrated excellent general performance [11]. We construct a Bayesian RF (BRF), via NPL with decision trees, under a prior mixing distribution (a variant of [S1]). This enables the incorporation of prior information, via synthetic data generated from a prior prediction function, in a principled way that is not available to RF. The step-like generative likelihood function arising from the tree partition structure does not reflect our beliefs about the true sampling distribution; the trees are just a convenient compression of the data. Because of this we simply update the prior in the MDP by specifying $\pi(\gamma|x_{1:n}) = \pi(\gamma)$. Details of our implementation of BRF can be found in Section 3 of the Supplementary Material.

To demonstrate the ability of BRF to incorporate prior information, we conducted the following experiment. For 13 binary classification datasets from the UCI ML repository [10], we constructed a prior, training and test dataset of equal size. We measured test dataset predictive accuracy for three methods (relative to an RF trained on the training dataset only): BRF (c=0) (a non-informative BRF with $c = 0$, trained on the training dataset only), BRF (c>0) (a BRF trained on the training dataset, incorporating prior pseudo-samples from a non-informative BRF trained on the prior dataset, setting $c$ equal to the number of observations in the prior dataset), and RF-all (an RF trained on the combined training and prior datasets). See Fig. 3 for boxplots of the test accuracy over 100 repetitions.

As detailed in Section 3 of the Supplementary Material, for small $c$ we find that BRF and RF have similar performance, but as $c$ increases, more weight is given to the externally-trained component and we find that BRF outperforms RF. The best performance of our BRF tends to occur when $c$ is set equal to the number of samples in the external training dataset, in line with our intuition of the role of $c$ as an effective sample size. Overall, the BRF accuracy is better than that of RF, and close to that of RF-all. BRF may have privacy benefits over RF-all as it only requires synthetic samples; both the original data and the model can be kept private.

### 3.4 Direct updating of utility functions [S3]: population median

We demonstrate direct inference for a functional of interest using the population median, under a misspecified univariate Gaussian model, as an example, where the parameter of interest is $\alpha_0 = \arg\min_\alpha \int |\alpha - x| dF_0(x)$, and an MDP prior centered at a $\mathcal{N}(\theta, 1)$ with prior $\pi(\theta) = \mathcal{N}(0, 10^2)$ and data generated from a skew-normal distribution. We use the posterior bootstrap to generate posterior samples that incorporate the prior model information with that from the data. Figure 4 presents histograms of posterior medians given a sample of 20 observations from a skew-normal distribution with mean 0, variance 1 and median approximately $-0.2$. The left-most histogram is sharply peaked at the sample median but does not have support outside of $(x_{\min}, x_{\max})$. As $c$ grows smoothness from the parametric model is introduced to a point where the normal location parameter is used.

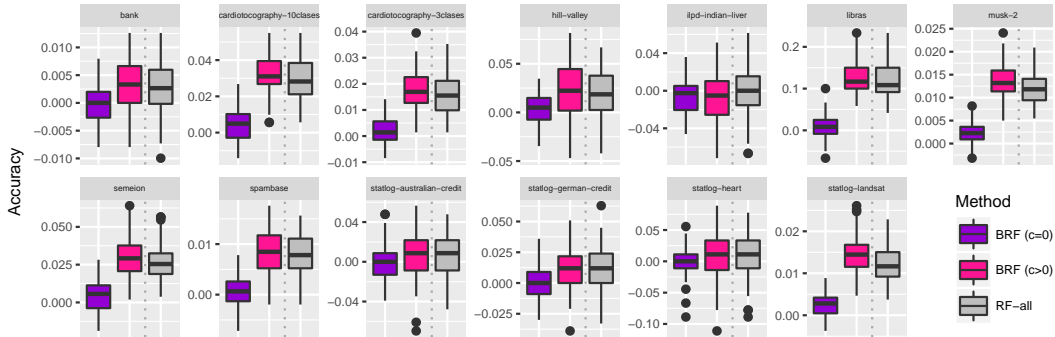

Figure 3: Box plot of classification accuracy minus that of RF, for 13 UCI datasets.

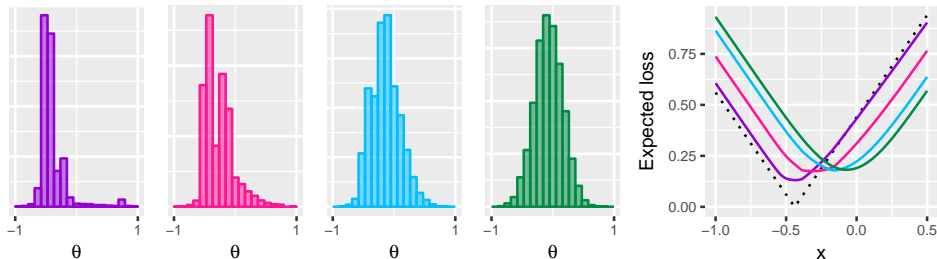

Figure 4: Posterior histogram for median (left to right) $c = 0, 20, 80, 1000$. Right-most: posterior expected loss as a function of observation $x$. Dotted line shows the loss to the sample median.

## 4   Discussion

We have introduced a new approach for scalable Bayesian nonparametric learning, NPL, for parametric models that facilitates prior regularization via a baseline model, and corrects for model misspecification by incorporating an empirical component that has greater influence as the number of observations grows. A concentration parameter $c$ encodes subjective beliefs on the validity of the model; $c = \infty$ recovers Bayesian updating under the baseline model, and $c = 0$ ignores the model entirely. Under regularity conditions, asymptotically, our method closely matches parametric Bayesian updating if the posited model were indeed true, and will provide an asymptotically superior predictive if the model is misspecified. The NP posterior predictive mixes over the parametric model space as opposed to targeting $F_0$ directly, though this may aid interpretation compared to fully nonparametric approaches. Additionally, our construction admits a trivially parallelizable sampler once the parametric posterior samples have been generated (or if $c = 0$).

Our approach can be seen to blur the boundaries between Bayesian and frequentist inference. Conventionally, the Bayesian approach conditions on data and treats the unknown parameter of interest as if it were a random variable with some prior on a known model class. The frequentist approach treats the object of inference as a fixed but unknown constant and characterizes uncertainty through the finite sample variability of an estimator targeting this value. Here we randomize an objective function (an estimator) according to a Bayesian nonparametric prior on the sampling distribution, leading to a quantification of subjective beliefs on the value that would be returned by the estimator under an infinite sample size.

At the heart of our approach is the notion of Bayesian updating via randomized objective functions through the posterior bootstrap. The posterior bootstrap acts on an augmented dataset, comprised of data and posterior pseudo-samples, under which randomized maximum likelihood estimators provide a well-motivated quantification of uncertainty while assuming little about the data-generating mechanism. The precursor to this is the weighted likelihood bootstrap, which utilized a simpler form of randomization that ignored prior information. Our approach provides scope for quantifying uncertainty for more general machine learning models by randomizing their objective functions suitably.

## Acknowledgements

SL is funded by the EPSRC OxWaSP CDT, through EP/L016710/1. CH gratefully acknowledges support for this research from the MRC, The Alan Turing Institute, and the Li Ka Shing foundation.

## Footnotes

[1]The MDP should not to be confused with the Dirichlet process mixture model (DPM) [18].

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
