[Supplementary Material · np_learning_supplementary_NeurIPS.pdf]

# Supplementary Material for "Nonparametric learning from Bayesian models with randomized objective functions"

**Simon Lyddon**
Department of Statistics
University of Oxford
Oxford, UK
lyddon@stats.ox.ac.uk

**Stephen Walker**
Department of Mathematics
University of Texas at Austin
Austin, TX
s.g.walker@math.utexas.edu

**Chris Holmes**
Department of Statistics
University of Oxford
Oxford, UK
cholmes@stats.ox.ac.uk

## 1   Setting the MDP concentration parameter

The MDP concentration parameter $c$ represents the subjective faith we have in the model: setting $c = 0$ means we discard the baseline model completely; whereas for $c = \infty$ we obtain the Bayesian posterior, equivalent to knowing that the model is true.

One way that the concentration parameter may be specified in practice is via the prior uncertainty of a functional. For example, we could set $c$ using a priori variance of the population mean. Using standard properties of the Dirichlet process, it is straightforward to show that under an $\mathrm{MDP}(\pi(\theta), c, f_\theta(\cdot))$, the variance in the mean functional is given by

$$\mathrm{var}\,\mu(\theta) \;+\; \frac{1}{1+c}\,\mathbb{E}\,\sigma^2(\theta)$$

where $\mu(\theta)$ is the mean of $x|\theta$ under the model, and $\sigma^2(\theta)$ is the variance of $x|\theta$, and the variance and expectation in the expression above are with respect to the prior $\theta \sim \pi$. Thus, if we can elicit a prior variance over the population mean, this will lead directly to a specific setting for $c$.

Another option is the Bayesian learning of $c$ via a hyper-prior; see [3], Section 4.5 for details. In practice it may be preferable to consider a number of difference values of $c$; once the parametric posterior samples are generated the posterior for each value of $c$ could be computed in parallel.

## 2   Posterior Bootstrap for adaptive Nonparametric Learning

We present the posterior bootstrap for adaptive Nonparametric Learning (aNPL), as discussed in Section 2.6 of the paper. This algorithm uses the stick-breaking construction of the Dirichlet process, as opposed to the Dirichlet distribution approximation of Section 2.5. The stick-breaking threshold, at which point the stick-breaking process ceases, is a function of the number of observations $n$ directly, meaning that for large enough $n$ the parametric model will be ignored, switching off the pseudo-samples from the parametric posterior predictive.

---
**Algorithm 1:** The Posterior Bootstrap for adaptive Nonparametric Learning
---
**Data:** Dataset $x_{1:n} = (x_1, \ldots, x_n)$.
Parameter of interest $\alpha_0 = \alpha(F_0) = \arg\max_\alpha \int u(x, \alpha) dF_0(x)$.
Mixing posterior $\pi(\gamma|x_{1:n})$, concentration parameter $c$, centering model $f_\gamma(x)$.
aNPL stick-breaking tolerance $\epsilon$.
**begin**
    **for** $i = 1, \ldots, B$ **do**
        Draw centering model parameter $\gamma^{(i)} \sim \pi(\gamma|x_{1:n})$;
        Draw model vs data weight: $s^{(i)} \sim \text{Beta}(c, n)$;
        Set data weights $w_{1:n}^{(i)} = (1 - s^{(i)})v_{1:n}^{(i)}$, with $v_{1:n}^{(i)} \sim \text{Dirichlet}(1, \ldots, 1)$;
        Set $v_{\text{rem}} = c/(c + n)$, $T = 0$;
        **while** $v_{rem} \geq \epsilon$ **do**
            $T \to T + 1$;
            Draw posterior pseudo-sample $x_{n+T}^{(i)} \sim f_{\gamma^{(i)}}$;
            Stick-break $w_{n+T}^{(i)} = s^{(i)} \prod_{j=1}^{T-1}(1 - v_{n+j}^{(i)})v_{n+T}^{(i)}$ with
            $v_{n+T}^{(i)} \sim \text{Beta}(1, c)$;
            Update $v_{\text{rem}} = c/(c + n) \prod_{j=1}^{T}(1 - v_{n+j}^{(i)})$;
        **end**
        Compute parameter update

$$\tilde{\alpha}^{(i)} = \arg\max_\alpha \left\{ \sum_{j=1}^{n} w_j^{(i)} u(x_j, \alpha) + \sum_{j=1}^{T} w_{n+j}^{(i)} u(x_{n+j}^{(i)}, \alpha) \right\};$$

    **end**
    Return NP posterior sample $\{\tilde{\alpha}^{(i)}\}_{i=1}^{B}$.
**end**

## 3  Bayesian Random Forests - further details

We base our Bayesian Random Forests (BRF) implementation on the Scikit-learn `RandomForestClassifier` class [4], with an alternative fitting method. Instead of applying bootstrap aggregation as RF does, for each tree we construct an augmented dataset, containing the training data and prior pseudo-data. We then fit the decision tree to the augmented dataset, suitably weighted.

If the concentration parameter $c$ is set to zero then no prior data is necessary; we just need to generate sample weights $w_{1:n} \sim \text{Dirichlet}(1, \ldots, 1)$ for observations $x_{1:n}$ and fit a tree to the weighted training dataset. When $c > 0$ we use a BRF with $c = 0$, trained on the prior data, to generate the prior pseudo-data. For each tree in the BRF we generate a pseudo-dataset before training a decision tree on the augmented dataset containing the prior pseudo-dataset and the training dataset. If $T$ pseudo-observations are generated, we generate a vector of $\text{Dirichlet}(1, \ldots, 1, c/T, \ldots, c/T)$ weights, relating to observations $x_{1:(n+T)}$, where $x_{1:n}$ is the training dataset and $x_{(n+1):(n+T)}$ is the prior pseudo-data.

When training RF decision trees, terminal node splits are made until the leaves only contain training data from a single class. This is computationally problematic for our BRF method, as after augmenting our internal training data with a large pseudo-dataset of samples, trees may need to grow ever deeper until leaf purity is attained. To avert this issue, we threshold the proportion of sample weight required to be present across samples at a node before a split can take place. This can be done via the `min_weight_fraction_leaf` argument of the Scikit-learn `RandomForestClassifier` class. In our testing we set a weight proportion threshold of $0.5(n + c)^{-1}$.

Internal, external and test data were obtained by equal sized class-stratified splits. Each forest contained 100 trees, and 10,000 external pseudo-samples were generated for the BRFs with $c > 0$. Predictions were made in our BRF by majority vote across the forest of trees.

A plot of the mean classification accuracy as a function of $c$ is given for the Bank dataset in Figure 1. As discussed in Section 3.3, for small $c$ the external data is given little weight and our method

performs similarly to an RF trained only on the internal data. As $c$ grows, the accuracy improves, peaking around where $c$ is equal to the number of external samples. This peak performance is roughly the same as that attained by an RF trained on both the external and internal data.

Figure 1: Mean classification accuracy vs $c$ for RF (black solid), RF-all (black dashed), and BRF (red) for varying $c$. over 100 repetitions, for the Bank dataset. Error bars for BRF and the grey lines for RF and RF-all represent one standard error from the mean. The BRF with $c = 0$ has a mean accuracy of 0.895 and a standard error of 0.0004.

Note that when setting $c = 0$ our method is equivalent to Bayesian bootstrapping random decision trees. [5] uses the Bayesian bootstrap as an underlying model of the data-generating mechanism, viewing the randomly weighted trees generated as a posterior sample over a functional of the unknown data-generating mechanism, similar to our construction. Previously, a number of attempts have been made in the literature to construct Bayesian models for decision trees [1, 2] but the associated MCMC sampling routines tend to mix poorly. Our method, whilst remaining honest that our trees are poor generative models, is very similar to RF in nature and performance. However, it has the additional benefit of enabling the user to incorporate prior information via a prediction function, in a principled manner.