[Reviews · NeurIPS 2018]

Reviewer 1



The idea: You want to do Bayesian inference on a parameter theta, with prior pi(theta) and parametric likelihood f_theta, but you're not sure if the likelihood is correctly specified. So put a nonparametric prior on the sampling distribution: a mixture of Dirichlet processes centered at f_theta with mixing distribution pi(theta). The concentration parameter of the DP provides a sliding scale between vanilla Bayesian inference (total confidence in the parametric model) and Bayesian bootstrap (no confidence at all, use the empirical distribution). This is a simple idea, but the paper presents it lucidly and compellingly, beginning with a diverse list of potential applications: the method may be viewed as regularization of a nonparametric Bayesian model towards a parametric one; as robustification of a parametric Bayesian model to misspecification; as a means of correcting a variational approximation; or as nonparametric decision theory, when the log-likelihood is swapped out for an arbitrary utility function. As for implementation, the procedure requires (1) sampling from the parametric Bayesian posterior distribution and (2) performing a p-dimensional maximization, where p is the dimension of theta. (1) is what you need to do in parametric Bayesian inference anyway, and (2) will often be a convex optimization problem, provided that your log-likelihood or your utility function is concave. You have to do (1) and (2) a bunch of times, but it's parallelizable. The illustrations of the method are wide-ranging and thoughtful. Section 3.2 shows that the method can be used to correct a variational Bayes logistic regression model; the corrected VB samples look like MCMC samples from the actual Bayesian posterior but are faster to generate than the MCMC samples. Section 3.3 explores how posterior predictive samples from this method could be passed as pseudo-data to an outside analyst who is not permitted to access the data directly. This is an interesting idea, though I'm not sure how it would compare to simple noise addition (or whatever other methods are used in the privacy literature). A quibble with the wording of lines 112-113 ("unlike MCMC, there are no issues or need for checking convergence..."), as the method does require sampling from the parametric Bayesian posterior, and this will often require checking convergence and using a burn-in phase, etc. In practice I think I'd struggle with choosing c, the DP concentration parameter, but when doing parametric Bayesian inference (c=infinity), it could be good practice to redo the inference under various finite values of c as a form of sensitivity analysis. The nice thing is that you can use the same set of posterior draws across all values of c, which is a real time-saver. In summary, this is a well-written paper that presents several potential applications of a simple, good idea. I recommend acceptance.

Reviewer 2



This paper extends Bayesian bootstrap ideas to optimizing a functional of some parameter with respect to an underlying but unknown distribution. Instead of simply generating random weights for the data set, the paper proposes additionally generating observations from a parametric centering distribution (with an empirical Bayes prior) and then generate random weights for the augmented data set based on a Polya urn scheme. The idea is that the augmentation acts as a regularizer, which combined with the random weights, gives a notion of uncertainty. The method is applied to generating more realistic VB posterior distributions and prior updates for Bayesian random forests. The method is tested on a small set of inference problems on UCI data sets. This paper is more confusing than it needs to be, and the large number of use cases does not help. While I really like the ideas in this paper, it is not quite where it needs to be for publication. My largest concerns center around tunable parameters: c, T (algorithm 1), and f_{\theta} (particularly when used as in algorithm 1). As we see in Figures 1 and 2, the method is highly sensitive to c. I am guessing that the same holds for T and especially f_{\theta}. Theoretical results on what these values should be, along with robust empirical demonstrations of the effects, would move my score into the accept/strong accept category. This line of research has a lot of promise where objective functions are generated from data and there is a large danger of overfitting to a small data sample (some RL, Bayesian optimization).

Reviewer 3



This paper proposes an interesting method to asymptotically achieve a better posterior predictive to the true model. The method itself can be seen as a intermediate between standard Bayesian inference and Bayesian bootstrap. The strategy is to integrate a Dirichlet process to the model, where likelihood shows up as the base measure. By tuning the total mass of the mean measure, we are able to control the 'objective level' of the entire inference. Even though not drawing from the true posterior, the method is asymptotically more efficient and it can parallel. I think the overall contribution of the paper is pretty good. But the writing could improve. For example [S1], the method is not solving the model misspecification problem, and the best it can achieve is still to find an optimal $f_\theta\in\mathcal{F}_0$ that minimize its KL divergence towards the true model. For [S2], since the authors do not show any non-asymptotic result about the gap between the posterior they derive and the true Bayesian posterior, I cannot quantity how good they 'approximate' the true posterior. For [S4], I don't think that is a main focus of the paper. The paper proposes a nice algorithm, but my concern is that sampling a DP posterior can be expensive. So I'm not sure whether the overall computational complexity is affordable. Overall this is a nice paper that balance a subjective and an objective view of Bayesian inference via a mixture of Dirichlet processes. My only concern is that the claimed points in the paper are not fully developed.